# A harmful religio-cultural practice (*Chhaupadi*) during menstruation among adolescent girls in Nepal: Prevalence and policies for eradication

**Dipendra S. Thakuri** [1]*, **Roshan K. Thapa**[2], **Samikshya Singh**[3], **Geha N. Khanal**[4], **Resham B. Khatri** [5,6]

**1** Health and Nutrition Department, Save the Children, Surkhet, Nepal, **2** Hilly Region Development Centre (HRDC), Jajarkot, Nepal, **3** Nepal Public Health Association, Kathmandu, Nepal, **4** Nepal Law Campus, Kathmandu, Nepal, **5** School of Public Health, Faculty of Medicine, University of Queensland, Brisbane, Australia, **6** Health Social Science and Development Research Institute, Kathmandu, Nepal

* dipendrathakuri95@gmail.com

**Data Availability Statement:** All relevant data are available within the paper and its Supporting information files.

## Abstract

### Background

*Chhaupadi* is a deeply rooted tradition and a centuries-old harmful religio-cultural practice. Chhaupadi is common in some parts of Karnali and Sudurpaschim Provinces of western Nepal, where women and girls are considered impure, unclean, and untouchable in the menstrual period or immediately following childbirth. In Chhaupadi practice, women and girls are isolated from a range of daily household chores, social events and forbidden from touching other people and objects. Chhaupadi tradition banishes women and girls into menstruation huts', or *Chhau huts* or livestock sheds to live and sleep. These practices are guided by existing harmful beliefs and practices in western Nepal, resulting in poor menstrual hygiene and poor physical and mental health outcomes. This study examined the magnitude of *Chhaupadi practice* and reviewed the existing policies for Chhaupadi eradication in Nepal.

### Methods

We used both quantitative survey and qualitative content analysis of the available policies. First, a quantitative cross-sectional survey assessed the prevalence of Chhaupadi among 221 adolescent girls in Mangalsen Municipality of Achham district. Second, the contents of prevailing policies on Chhaupadi eradication were analysed qualitatively using the policy cube framework.

### Results

The current survey revealed that most adolescent girls (84%) practised Chhaupadi in their most recent menstruation. The Chhaupadi practice was high if the girls were aged 15–17 years, born to an illiterate mother, and belonged to a nuclear family. Out of the girls

**Funding:** The authors didn't receive any funding support for this work.

**Competing interests:** The authors have declared that no competing interests exist.

practising Chhaupadi, most (86%) reported social and household activities restrictions. The policy content analysis of identified higher-level policy documents (constitution, acts, and regulations) have provisioned financial resources, ensured independent monitoring mechanisms, and had judiciary remedial measures. However, middle (policies and plans) and lower-level (directives) documents lacked adequate budgetary commitment and independent monitoring mechanisms.

## Conclusion

Chhaupadi remains prevalent in western Nepal and has several impacts to the health of adolescent girls. Existing policy mechanisms lack multilevel (individual, family, community, subnational and national) interventions, including financial and monitoring systems for Chhaupadi eradication. Eradicating Chhaupadi practice requires a robust multilevel implementation mechanism at the national and sub-national levels, including adequate financing and accountable systems up to the community level.

## Introduction

Menstruation is a normal biological process that indicates girls' entrance into womanhood [1]. However, despite being a natural phenomenon, it is perceived as a stigma and taboo in many parts of Nepal, considering women and girls impure and untouchable during the menstruation period [2, 3]. Furthermore, most women and girls in western hills (Karnali and Sudurpaschim provinces) are banished outside their homes to a makeshift hut or cowshed during their menstrual period [4]. Those small huts made up of mud and stones without windows and locks are called Chhaupadi huts, and this tradition is called Chhaupadi [5].

Chhaupadi is a centuries-old harmful practice guided by religio-cultural beliefs in western Nepal [2, 4]. The word Chhaupadi is derived from a local *Raute* dialect in the far-west where *"Chhau"* means untouchable or unclean, and *"Padi"* means being or becoming [6–8]. Thus, Chhaupadi refers to a state of being untouchable/unclean. There are several sociocultural taboos during the menstruation period in the western hills of Nepal [1, 7]. For instance, people believe women are impure, untouchable, and unclean during routine menstrual periods. Those societal beliefs limit women/girls involvement in daily activities, including restrictions in eating (milk and dairy products), touching (men, water sources, livestock, plants, and kitchen items) and visiting public places (water sources, temple, prayer room and cultural ceremonies) [5, 9]. Moreover, women and girls are forced to isolate themselves and sleep inside the menstrual hut or in the cattle sheds where health hygiene is largely compromised. However, the Chhaupadi practice is not limited to menstruation and is common during postpartum [10].

Women and girls face physical and mental hardship while residing in Chhaupadi huts during mensuration [7–9]. In the past, several Chhaupadi related incidents have been reported from western hills of Nepal, such as sexual abuse, rape, attack from wild animals, snake or scorpion bites, and illness. All of these were related to poor safety and unhygienic conditions [4, 11–13]. In addition, poor hygiene and sanitation practices in Chhaupadi are compounded by the unavailability of and poor access to water and sanitation facilities, lack of sanitary napkins, and healthy environmental conditions [4]. The poor state of menstrual hygiene results in adverse health outcomes such as reproductive and genitourinary tract infections, the risk of cervical cancer, anxiety and depression [14–16]. Furthermore, in many cases, Chhaupadi

practices have resulted in menstruation-related shame, fear and humiliation, poor menstrual hygiene practices, and girls' school absenteeism in Nepal [17–19] and other low and middle-income countries [1, 14, 17].

In Nepal, a high prevalence of Chhaupadi practice is reported in the western region. The multi-indicator cluster survey (MICS) 2019 reported 21.1% of women to practice Chhaupadi in Sudurpaschim province, which is six-fold higher than the national average (3.8%). This finding resonates with other sub-national small sample-sized studies [1, 10]. For instance, the prevalence of Chhaupadi in Achham district in 2011 was 95% among the women [10], while it was 72% among the adolescents' girls in 2018 [1]. Several sociocultural and religious factors are the major drivers behind it include illiteracy, superstitions, stigma, existing gender-based discrimination, cultural, traditional, and religious beliefs and the poor implementation of laws against Chhaupadi [5]. In addition, Chhaupadi has demonstrated inter-generational practice in Nepal [8], which violates Nepal's commitment to several national and international conventions to protect women's reproductive rights and the right to live without discrimination [5].

Many policies have been formulated and implemented targeting the eradication of Chhaupadi in Nepal. For instance, the Constitution of Nepal (2015) ensures the right to equality (Article 18) and the right to reproductive health (Article 38). Likewise, Article 24 (1) and Article 29 (2) affirm that "no one shall be treated with any kinds of untouchability or discrimination and no one shall be exploited based on any custom, tradition, culture, and practices or any other bases" [20]. In May 2005, the Supreme Court outlawed Chhaupadi as malpractice and directed the government to take necessary legal arrangements to eliminate Chhaupadi. In 2008, the Government of Nepal formulated the directives to eliminate the Chhaupadi practice [5]. Thereafter, many awareness-raising programs were focused on community stakeholders, and Chhaupadi sheds demolition drive has been implemented [8], aiming to eradicate Chhaupadi [21]. Recently, the Criminal Code Act (2017) criminalised Chhaupadi and included the provision of a three-month jail sentence and/or Nepalese Rupees (NRs) 3,000 (~USD 26) fine for anyone forcing a woman to follow the custom [22].

Despite Chhaupadi being a widespread harmful religio-cultural practice in western Nepal, limited studies have identified the magnitude of Chhaupadi practice among adolescent girls [1, 23]. Furthermore, there is a lacking systematic mapping of contents in existing legal and policy documents related to Chhaupadi eradication. Chhaupadi is a problem rooted in sociocultural and religious values. Therefore, it is imperative to review the content of existing policies and explore the multiple dimensions of policy loopholes to strengthen their implementation in federalised governance systems of Nepal. The findings can inform policymakers and program managers to design, revise and implement tailored Chhaupadi eradication interventions at the federal, provincial, and community levels.

## Methods

We used quantitative and qualitative methods to identify the magnitude of Chhaupadi and to understand the policy landscape on Chhaupadi, respectively [Fig 1]. The purpose of the quantitative survey was to identify the recent extent of the Chhaupadi practice, while the qualitative method was used to complement and expand the scope of the research gaps [24]. Thus, by reviewing a past study [25], we used multiple methods to identify the recent prevalence of Chhaupadi practice and eradication policy initiatives. Second, we conducted a cross-sectional survey using face-to-face interviews among adolescent girls. Third, we revised the content of existing policies with a focus on Chhaupadi eradication. The rationale for using policy cube was to provide a bigger perspective of the problem. Finally, we integrated the quantitative and

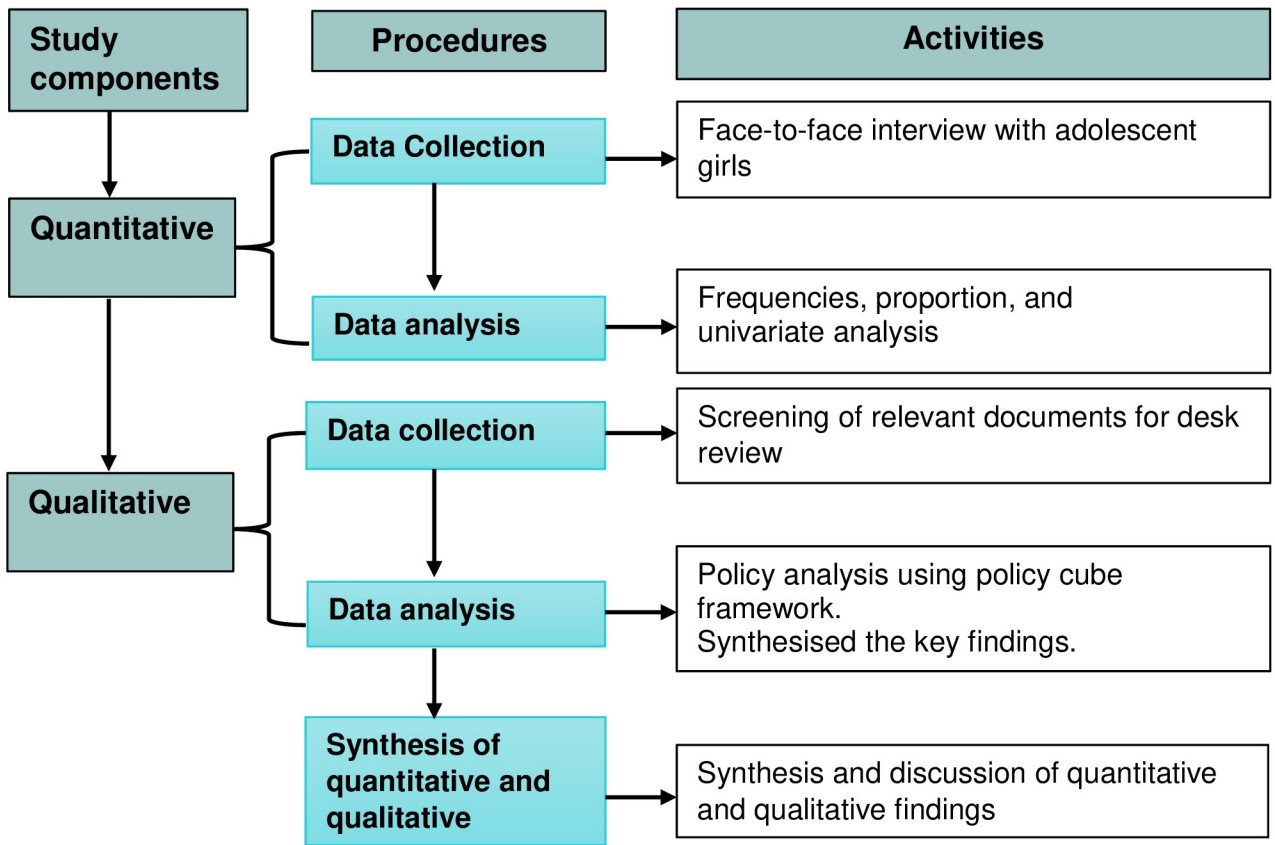

**Fig 1. Flow chart of quantitative and qualitative components of the study.**

qualitative methods at the conceptual level. Finally, we discussed the quantitative and qualitative findings to draw the policy/program and research implications.

## Quantitative component of the study

**Study design, setting and sampling.**   A cross-sectional study was conducted among adolescent girls aged 11 to 19 years. Interviews were conducted between 15 May and 15 Sep 2017. The adolescence period is the critical stage of habit formation, and the majority of adolescents' girls experience puberty and the first mensuration in their early adolescence. Moreover, adolescent girls usually follow what parents and society instruct them to do on their health and hygiene, which could shape their behaviour; therefore, we decided to recruit adolescent girls in this study.

As available literature suggests, Chhaupadi is widely prevalent in the Karnali and Sudurpaschim provinces of Nepal [1, 5, 26]. Achham is one of the districts in Sudurpaschim province where many incidents, including deaths in Chhau huts, were reported in media over the last decade [1, 10, 27]. We selected Achham as a study district, which had 11 rural local government units. Mangalsen municipality was purposively selected considering feasibility, time factor, accommodation, and available resources for data collection. According to the National Census of 2011, Mangalsen municipality had 6,604 households with about 32,507 population, of which 8,265 were adolescents [28]. Two wards (wards no 6 and 7) of Mangalsen

municipality were randomly selected. The sampling frame was the population of adolescent girls aged 11–19 years. The list of households with adolescent girls was accessed from the respective ward office. There were about 650 adolescent girls aged 11–19 years in study wards.

We calculated the sample size based on the prevalence of Chhaupadi practice (15%) reported in MICS 2014 [26]. We assumed an allowable error of 5% and a non-response rate of 10% for calculating the sample size. We used the formula N = Z2pq/d2 [where Z = 1.96, p = 0.15, q = 0.85), d = 0.05] and sample size was estimated to be 221. We selected 221 households of adolescent girls through a simple random sampling technique (every fourth household). An adjoining household was recruited if a girl was absent in a selected household at the survey time. Additionally, in a case where one household had more than one eligible adolescent girl, the eldest adolescent girl was selected for the interview.

**Study variables.** The respondents were asked whether they practised Chhaupadi in their most recent menstrual period prior to the survey. This outcome variable had the response of either 'yes' or 'no'. Explanatory variables were selected based on the review of previous literature [1, 2, 29]; we selected sociodemographic, knowledge and practices related variables. Sociodemographic related characteristics of girls included ethnicity (advantaged: Brahmin and Chhetri, and disadvantaged: Dalits and Janajati) [30], respondent's age (11–14, 15–17, and 18–19 years), respondent's education (6–8 grade and 9–12 grade), education of parents (illiterate or no education, basic education (1–8 grade), and secondary (9–10 grade) or above), occupation of mother (unpaid work and paid cash work), occupation of father (job, agriculture, business, and labor work) and the family type (nuclear and joint). At the same time, variables related to knowledge and practice on Chhaupadi included knowing about Chhaupadi (yes, no), factors for Chhaupadi practice (impurity, fear of God, inadequate knowledge, societal fear, and fear of family members falling sick) and the person suggesting staying in Chhaupadi huts (parents, grandparents, and relatives or traditional healers). We further categorised restriction during menstruation: bathing restriction (yes, no), food restriction (yes, no), types of food restriction included (milk and dairy products, vegetable and fruits, and meat and meat products), restriction of activities (yes, no), types of restricted activities (entering the kitchen or cooking, sitting, or touching or seeing male members, wearing new clothes, and buying or touching medicine). Menstrual hygiene was categorised based on place of bathing: home tap, *chhau-dhara* (designated taps for those practising Chhaupadi), *chhau kuwa* (designated well for those practising Chhaupadi), and river. Menstrual hygiene management practice included using a sanitary pad (yes, no), reasons for not using a sanitary pad (costly or unavailability, disposal, and other problems). Similarly, sanitary pad related variables included types of absorbents (sanitary pad, new clothes, home-made sanitary pad, and old washed clothes) and places to store sanitary absorbents (own room, cowshed, and wrapped with other clothes).

**Data collection and analysis.** The survey tool was developed based on previous literature [1, 2, 14, 29, 31]. For quality assurance, survey tool was pretested among 20 adolescent girls of the adjoining ward (ward no 8) within Mangalsen municipality. Necessary modifications to the survey tool were made mainly in the flow of the pattern of questions and language. The second author (RKT) collected data using the revised questionnaire, and interviews were conducted in the Nepali language. Collected data were entered in Microsoft Excel first, and then necessary data validations and cross-checks were made to avoid possible errors during data entry. Data was exported to Stata (version 14.1) (Stata Crop, Texas, USA). Findings of descriptive analysis were reported with frequency and proportion of Chhaupadi practice. We further estimated the association between outcome and explanatory variables using the Chi-square test.

## Research ethics

Ethical approval was obtained from the Institutional Review Board (IRB) of Chitwan Medical College (CMC), Tribhuvan University. The Nepal Health Research Council (NHRC) generally delegates ethical approval authority to academic institutions to provide ethical approval for institutional research [32]. Furthermore, permission was taken from Mangalsen municipality and Achham District Health Office. We obtained written informed consent from the respondents prior to the interview. Participation of the study respondents was voluntary where the respondents could refuse the interview at any time. Any kind of support, including financial incentives, was not provided to the respondents. Furthermore, verbal parental consent was obtained for adolescents below 15 years.

## Qualitative component of the study

The qualitative component of the study constituted the policy content review process using the policy-cube framework developed by Buse and colleagues [33]. Initially, the policy cube framework was developed to understand the strength of national policies to combat and prevent diet-related Non-Communicable Diseases (NCDs) [33]. Then, all the relevant policy documents (constitution, laws, regulations, policies, strategies, directives, judicial orders) were then identified and reviewed, focusing on Chhaupadi eradication. Then, the relevant policy text was extracted and fitted into the framework.

**Identification and selection of policy documents.** The detail of the relevant policy selection process is presented in Fig 2. We identified relevant policies through web search and consultation with experts involved in the Chhaupadi eradication movement in Nepal. We searched different websites to identify the available policies in Chhaupadi eradication in Nepal. Keywords in web search included Karnali and Sudurpaschim province, harmful Chhaupadi practice, religio-cultural practice, practice, Chhau huts, Chhaupadi eradication, Chhaupadi practice, menstrual hygiene policy and Nepal. The documents search was conducted in June 2020. In addition, the authors (RBK, GNK, DST) consulted with experts having relevant experiences in Chhaupadi eradication and requested them to suggest the relevant additional policies. First, we identified policy documents with components of menstrual hygiene management, reproductive health and Chhaupadi practice. Second, we looked at Chhaupadi specific contents in Constitution, Acts, Regulations, policy, directives, and plans of three layers of governments (federal, provincial, and local).

**Framework for qualitative data analysis.** We used a policy cube framework that comprises three dimensions i) comprehensiveness, ii) political salience with effective means of implementation, and iii) principles of equity and human rights. Comprehensiveness assesses to what extent the existing policies have covered the components to eradicate Chhaupadi practice. First, we reviewed the content of the federal, provincial, and local level policy documents on Chhaupadi eradication. Second, political salience and effectiveness of means of implementation examined policy authority level, budget line item, and accountability system. We mapped the policies by their level of authority and categorised them into highest, middle, and lowest levels. The highest-level policy documents are comprised of the constitution, national laws, and regulations. Policies, strategies, rules, and action plans formulated by federal, provincial, and local levels were categorised as middle-level documents, while directives, guidelines, action plans and implementation plans are considered the documents with the lowest level of authority [33].

Furthermore, we assessed whether budgetary commitments were stated in different levels of policy documents. We also looked at accountability mechanisms by identifying implementing agencies, independent monitoring systems, and remedial actions for policy non-

<div style="border: 1px solid #000;">

**Strategies for selection and identification of policy and program documents**

Web searching and coordinating with relevant stakeholders (Governmental and Non-Governmental)

</div>

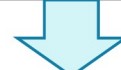

<div style="border: 1px solid #000;">

**Document identified or available**

Constitution-1, Acts- 5, National Policy-5, Provincial Policy-1, Municipal Policy and Plan-1, Judicial order-1, Directives-1, Regulation-1, Strategy-3

</div>

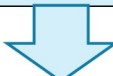

<div style="border: 1px solid #000;">

**Policy and program document selection**

Documents which are relevant to Chhaupadi eradication and that describes key policy, strategies, and program actions, published in Nepali or English language

</div>

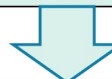

<div style="border: 1px solid #000;">

**Policy and program documents included for final review**

Total 8 policy and program documents were included for final review

</div>

**Fig 2. Flowchart of selection of policy and program documents on Chhaupadi eradication in Nepal.**

compliance. Third, the principle of equity and human rights dimension assesses the policies to what extent the principle of human rights and social equity and justice were articulated. Finally, we evaluated whether these documents ensure the right to live with self-dignity and self-determination.

## Results

### Prevalence of Chhaupadi practice

Table 1 shows the sociodemographic profile and magnitude of Chhaupadi among adolescent girls in Mangalsen municipality. Out of total adolescent girls (N = 221), most (84%) practised Chhaupadi during their last menstruation. Over half (56.1%) of the girls were aged between 15 and17 years. Three-fourth of them were from advantaged ethnic groups. More than half (58.8%) had completed secondary education. Four in five (81%) respondents' mothers were unpaid workers. A substantial proportion (93.5%, n = 124) of girls who practised Chhaupadi were aged 15–17 years. The daughters of illiterate mothers practised Chhaupadi more (90% of n = 126) than mothers with a secondary level of education (75%, n = 24). The practice was

**Table 1. Chhaupadi practice stratified by sociodemographic variables among adolescent girls aged 11–19 years in Mangalsen municipality, Achham district, 2017.**

| Variables (N = 221) | Total | Chhaupadi | | P-value |
|---|---|---|---|---|
| | n (%) * | No (%) # | Yes (%) # | |
| **Age (years)** | | | | |
| 11–14 | 76 (34.4) | 22 (29.9) | 54 (71.1) | 0.001† |
| 15–17 | 124 (56.1) | 8 (6.5) | 116 (93.5) | |
| 18–19 | 21 (9.5) | 5 (23.8) | 16 (76.2) | |
| **Ethnicity** | | | | |
| Disadvantaged ethnicities | 54 (24.4) | 5 (9.3) | 49 (90.7) | 0.128 |
| Advantaged ethnicities | 167 (75.6) | 30 (18.0) | 137 (82.0) | |
| **Education** | | | | |
| Basic (class 6–8) | 91 (41.2) | 19 (20.9) | 72 (79.1) | 0.086 |
| Secondary (class 9–12) | 130 (58.8) | 16 (12.3) | 114 (87.7) | |
| **Maternal education** | | | | |
| Illiterate | 126 (57.0) | 13 (10.3) | 113 (89.7) | 0.034† |
| Primary | 71 (32.1) | 16 (22.5) | 55 (77.5) | |
| Secondary and higher | 24 (10.9) | 6 (25.0) | 18 (75.0) | |
| **Paternal education** | | | | |
| Illiterate | 45 (20.4) | 3 (6.7) | 42 (93.3) | 0.090 |
| Primary | 108 (48.9) | 17 (15.7) | 91 (84.3) | |
| Secondary and higher | 68 (30.8) | 15 (22.1) | 53 (77.9) | |
| **Mother's occupation** | | | | |
| Unpaid | 179 (81.0) | 27 (15.1) | 152(84.9) | 0.527 |
| Paid cash work | 42 (19.0) | 8 (19.0) | 34 (81.0) | |
| **Father's occupation** | | | | |
| Job | 38 (17.2) | 6 (15.8) | 32 (84.2) | 0.410 |
| Agriculture | 127 (57.5) | 17 (13.4) | 110 (86.6) | |
| Business | 31 (14.0) | 8 (25.8) | 23 (74.2) | |
| Labor | 25 (11.3) | 4 (16.0) | 21(84.0) | |
| **Family type** | | | | |
| Nuclear | 133 (60.2) | 15 (11.3) | 118 (88.7) | 0.022† |
| Joint | 88 (39.8) | 20 (22.7) | 68 (77.3) | |

P value obtained from the chi-squared test of association.

† Significant at p < 0.05.

* Column percentage.

# Row percentage.

higher among the girls living in a nuclear family (89%, n = 133) compared to joint family (77%, n = 88) (Table 1).

Table 2 shows the Chhaupadi related knowledge, restrictions, and hygiene practices among adolescent girls in their last menstrual period. Most (98.2%) girls knew about Chhaupadi. Four in five (84.2%) girls faced restrictions in foods. Likewise, over nine in ten (94.1%) girls faced restrictions in social and household activities. More than two-thirds (70.1%) of girls did not use sanitary pads, while over two-fifth (44.3%) of adolescent girls used new cloth in their last menstrual period. Nearly three quarter (73.5%) of the girls reported high cost and unavailability of the sanitary pad as a reason for not using it. However, the majority (84.8%) of girls who knew about Chhaupadi practised it. Most girls who practised Chhaupadi (86.5% of n = 208) faced social and household level restrictions (Table 2).

**Table 2. Knowledge, restrictions, and hygiene practice among adolescent girls in their most recent menstrual period in Mangalsen municipality, Achham district, 2017.**

| Variables | Total (%) * | Chhaupadi | | P-value |
|---|---|---|---|---|
| | | Yes (%) # | No (%) # | |
| **Knowledge about Chhaupadi** | | | | |
| **Knows about Chhaupadi** | | | | |
| Yes | 217 (98.2) | 184 (84.8) | 33 (15.2) | 0.001† |
| No | 4 (1.8) | 1 (25.0) | 3 (75.0) | |
| **Factors associated with Chhaupadi** | | | | |
| Impurity | 47 (21.3) | 43 (91.5) | 4 (8.5) | 0.430 |
| God angry | 47 (21.3) | 40 (85.1) | 7 (14.9) | |
| Inadequate knowledge | 79 (35.7) | 66 (83.5) | 13 (16.5) | |
| Societal fear | 16 (7.2) | 12 (75.0) | 4 (25.0) | |
| Sick family members | 32 (14.5) | 25 (78.1) | 7 (21.9) | |
| **Person suggesting staying in Chhaupadi huts** | | | | |
| Parents | 88 (39.8) | 73 (83.0) | 15 (17.0) | 0.917 |
| Grandparents | 88 (39.8) | 75 (85.2) | 13 (14.8) | |
| Relatives/traditional healers | 45 (20.4) | 38 (84.4) | 7 (15.6) | |
| **Restriction during menstruation** | | | | |
| **Bathing in public sources of water** | | | | |
| No | 82 (37.1) | 66 (80.5) | 16 (19.5) | 0.250 |
| Yes | 139 (62.9) | 120 (86.3) | 19 (13.7) | |
| **Food restriction** | | | | |
| Yes | 186 (84.2) | 158 (84.9) | 28 (15.1) | 0.050† |
| No | 35 (15.8) | 28 (80.0) | 7 (20.0) | |
| **Restricted food types** | | | | |
| Milk and dairy product | 166 (89.2) | 142 (85.5) | 24 (14.5) | 0.018† |
| Vegetable/ fruits | 9 (4.8) | 5 (55.6) | 4 (44.4) | |
| Meat/meat product | 11 (5.9) | 11 (100.0) | 0 (0.0) | |
| **Activity restriction** | | | | |
| Yes | 208 (94.1) | 178 (85.6) | 30 (14.4) | 0.021† |
| No | 13 (5.9) | 8 (61.5) | 5 (38.5) | |
| **Activity-related restriction** | | | | |
| Entering kitchen/cooking | 167 (80.3) | 143 (85.6) | 24 (14.4) | 0.050† |
| Sitting/touching/seeing male member | 36 (17.3) | 32 (88.9) | 4 (11.1) | |
| Wearing new cloth and buying or touching medicines | 5 (2.4) | 3 (60.0) | 2 (40.0) | |
| **Bathing places** | | | | |
| Home tap | 51 (23.1) | 38 (74.5) | 13 (25.5) | 0.174 |
| *Chhau-dhara* | 115(52.0) | 99 (86.1) | 16 (13.9) | |
| *Chhau kuwa* | 24 (10.9) | 21 (87.5) | 3 (12.5) | |
| River | 31 (14.0) | 28 (90.3) | 3 (9.7) | |
| **Menstrual hygiene** | | | | |
| **Using sanitary pads** | | | | |
| Yes | 66 (29.9) | 47 (71.2) | 19 (28.8) | 0.001† |
| No | 155 (70.1) | 139 (89.7) | 16 (11.3) | |
| **Reasons for not using sanitary pads** | | | | |
| Costly/unavailable | 114 (73.5) | 106 (93.0) | 8 (7.0) | 0.024† |
| Disposal and other problems | 41 (26.5) | 33 (80.5) | 8 (19.5) | |
| **Types of absorbent used** | | | | |

*(Continued)*

**Table 2.** (Continued)

| Variables | Total (%) * | Chhaupadi | | P-value |
|---|---|---|---|---|
| | | Yes (%) # | No (%) # | |
| Sanitary pad | 28 (12.7) | 17 (60.7) | 11 (39.3) | 0.004† |
| New cloth | 98 (44.3) | 87 (88.8) | 11 (11.2) | |
| Homemade sanitary pad | 62 (28.1) | 53 (85.5) | 9 (14.5) | |
| Old wash cloth | 33 (14.9) | 29 (87.9) | 4 (12.1) | |
| **Places to store absorbent** | | | | |
| In own room | 133 (60.2) | 113 (85.0) | 20 (15.0) | 0.922 |
| Cowshed | 18 (8.1) | 15 (83.3) | 3 (16.7) | |
| With other clothes | 70 (31.7) | 58 (82.9) | 12 (17.1) | |

P-value obtained from the chi-squared test of association.

† Significant at p < 0.05.

*Column percentage.

#Row percentage.

## Chhaupadi eradication policies in Nepal

We reviewed the federal, provincial, and local level policy documents (constitution, laws, regulations, policies, plans, strategies, directives, judicial orders), focusing on the contents of Chhaupadi eradication. The chronological policy trajectory of Chhaupadi eradication is shown in S1 Fig. Findings of policy review explained in three dimensions of policy cube framework: i) comprehensiveness, ii) salience features and effectiveness of means of implementation, and iii) principle of equity and rights. The first and third dimensions, we explained narratively, and the second is presented in Table 3.

## Comprehensiveness of policies

There are some health and social policies in Nepal on women's health and reproductive rights; however, they lack an implicit explanation of Chhaupadi eradication strategies. For instance, the Safe Motherhood and Reproductive Health Rights (SMRHR) Act (2018) has not mentioned the eradication of Chhaupadi practice. Although the Local Government Operation Act (LGOA) (2017) has mentioned ward level jurisdiction on eradicating harmful practices like Chhaupadi, there are no procedural laws, including regulations that delineate the eradication interventions. The Criminal (Code) Act (2017) criminalises the accuser (who promotes the Chhaupadi custom) and imprisons and/or penalizes [22]. Contrarily, the first two years of implementation experience has revealed enervated legal execution [23].

Similarly, some policies lack content and execution and inadequate coherence in constitutional mandates and sectoral policies. For instance, National Health Policy (2019) [34] has not specified any provisions to menstrual healthcare services. The lack of such health policy provision contradicts the constitutional commitment of providing free essential health services for all citizens. Similarly, evidence shows that menstrual restrictions are factors for poor learning achievements and education outcomes among girls and women [35]. However, National Education Policy (2019) [36] has not anticipated any challenges girls face in accessing education during such restriction periods. Likewise, although National Youth Policy (2010) aims to render necessary support to vulnerable youths [37], the policy has not anticipated the issues faced by youths like inferiority complex, deprivation from participation and loss of dignity when girls are forced to follow Chhaupadi practice [35]. In addition, the Mental Health Policy

**Table 3. Salience features and effectiveness of means of implementation of relevant policies on the eradication of Chhaupadi practice in Nepal, 2020.**

| Level of Authority | Policy Document | Description | Provision related to Chhaupadi | Implementing agency | Budgetary Commitment | Accountability System | | |
|---|---|---|---|---|---|---|---|---|
| | | | | | | Reporting in Public Domain | Independent monitoring system | Remedial Action/ Mechanism |
| Highest level | Constitution of Nepal (Government of Nepal, 2015) | Fundamental law of the country that guarantees the following fundamental rights: the right to live with dignity, right against exploitation, right to equality, rights of women, right to freedom, right relating to food, rights relating to clean environment and rights relating to appropriate housing. | Constitutionally guaranteed fundamental rights are violated by Chhaupadi customs. | Council of Ministers | Yes | NHRC | NHRC | Constitutional remedy against the violation of fundamental rights |
| | Right to Safe Motherhood and Reproductive Health Act, 2018 | Making necessary provisions on motherhood and reproductive health service safe and fulfil the right to safe motherhood and reproductive health of the women conferred by the Constitution of Nepal. | Legislative provision to provide education, information, counselling, and service relating to sexual and reproductive health to every woman and teenagers. | MoHP, LGs | Yes | Not mentioned | NWC | Judicial remedy |
| | Criminal (Code) Act, 2017 (Government of Nepal, 2017a) | Amendment and consolidation of the existing laws related to criminal offences and maintaining law and order in the country. | Legislative provision of punishment with up to three months imprisonment and/or NPR 3,000 ($26) fine for those forcing a recently delivered or menstruating woman to stay in a menstrual hut. | Ministry of Home Affairs | Yes | NHRC | NHRC, NWC | Judicial remedy |
| | Local Government Operation Act 2017(Government of Nepal, 2017b) | Legal provision regarding the legislative, executive, and judicial practice of local governments. | Legislative provision to ensuring the basic healthcare services, including reproductive healthcare under the jurisdiction of local government. | LGs | Yes | Not mentioned | | Judicial remedy |
| | Domestic Violence (Offence and Punishment) Act, 2009 (Government of Nepal, 2009) | Legal provision respecting the right of every person to live in a secure and dignified life and providing protection to the victim of violence. | Legislative provision of punishment with a fine of up to 25,000 (US$ 215) and/or six months of imprisonment for those who commits any acts of domestic violence. | Ministry of Home Affairs | Yes | NHRC | NHRC, NWC | Judicial remedy |

*(Continued)*

**Table 3.** (Continued)

| Level of Authority | Policy Document | Description | Provision related to Chhaupadi | Implementing agency | Budgetary Commitment | Accountability System | | |
|---|---|---|---|---|---|---|---|---|
| | | | | | | **Reporting in Public Domain** | **Independent monitoring system** | **Remedial Action/ Mechanism** |
| **Middle level** | *Chhaupadi* malpractice elimination Policy, 2019 (Sudurpaschim Province) | Specific policy related to the elimination of *Chhaupadi* practice was formulated by the provincial government. | Awareness and education interventions to change the *Chhaupadi* customs and ensure the rights of women, their protection and compensation through school education, mobilisation of several stakeholders including traditional healers, local leaders, priests, and local level networks.<br><br>Eliminate the *Chhaupadi* malpractices, superstition, custom and mindset through discouraging it by appropriate legal actions, and punishment measures that encourage or practice *Chhaupadi*. | Provincial, district, municipal and ward level implementation mechanisms are ensured | Directed the provincial and local government to allocate resources | Not available | Not available | Not available |
| | Municipal Policy and Plan (Mangalsen Municipality, 2019) | Local government plan and policy | Commitment to implement education and awareness campaigns in menstrual management and menstrual hut-free municipality | Local Government | No secured budgetary line-item allocation | Not available | Not available | Not available |
| **Low Level** | Chhaupadi Practice Elimination Directives, 2008 | The directive formulated after the judicial order of 2005 | Had immediate interventions like public awareness activities, health education and long-term interventions like women empowerment through proportional participation of women and gradual legal reforms to ensure human rights of women | Multisectoral agencies in the district and local levels | No secured budgetary allocation | Not available | Not available | Not available |

NHRC: National Human Right Commission.

NWC: National Women Commission.

MoHP: Ministry of Health and Population.

LGs: Local Governments.

(1996) has not provisioned any psychological problems as a consequence of Chhaupadi practice [38].

## Salience features and effectiveness of means of implementation

Table 3 depicts the contents of multilevel (highest, middle, and lowest levels) policy documents with their salience features, implementing agency, budgetary commitments, accountability system, independent monitoring, and remedial mechanisms. The highest-level policy document (constitution, laws, and regulations) must go through the parliamentary approval processes. Such legislative processes have strong legitimacy to influence, induce and/or enforce policy compliance [33]. Consequently, the federal, provincial, and local governments formulate policies, plans and strategies to operationalise the parliament-approved higher-level legal documents. In addition, the lowest level policies constitute directives, strategies, and action plan that neither has robust legislative compliance nor adequate implementation mechanism.

**Highest level policy documents (e.g., constitutions, acts and regulations).** The highest level policy documents comprise the constitution, laws, and regulations usually endorsed by the parliament [33]. For example, the constitution of Nepal (2015) commits to prohibit any kind of violence or exploitation on the grounds of religion, social, cultural tradition, or practices. Several national laws have been formulated with this mandate. The national law that had the provision of Chhaupadi practice is the Criminal (Code) Act (2017). This Act has the provision of three months' imprisonment or a financial penalty of NRs 3,000 (~USD 26) for those forcing a recently delivered or menstruating woman to stay in Chhaupadi huts [22].

Some Acts incorporate Chhaupadi eradication related provisions indirectly, such as the Domestic Violence (Offence and Punishment) Act (2009) [39] and Local Government Operation Act (LGOA) (2017) [40]. The former has the legal provision of imprisonment for any acts of domestic violence, like forcing the girls to stay in Chhaupadi huts. At the same time, the latter ensures the local government's jurisdiction to implement the activities to eliminate Chhaupadi practice and formulate appropriate legislative measures to provide essential healthcare services. Additionally, the Safe Motherhood and Reproductive Health Rights (SMRHR) Act (2018) has a legislative provision to ensure access to fundamental rights (education, information, counselling, and health services, including reproductive healthcare) for every woman [41]. However, policies have not specified any provisions for the management of menstrual hygiene and Chhaupadi practice.

**Middle-level policy documents (e.g., policies, plans and strategies).** The middle-level policy documents consist of policies, plans and strategies formulated by federal, provincial, and local governments. National Dignified Menstruation Policy (2020) is one of them, which has been drafted and yet to be endorsed [42]. On the other hand, there are some policy efforts from provincial and local governments. For instance, the Chhaupadi Malpractice Elimination Policy (2019) was endorsed by Sudurpaschim Province [43]. Likewise, Mangalsen Municipality endorses an annual work plan and budget (2019) incorporating the provisions to eradicate Chhaupadi practice [44–46].

All these documents have considered Chhaupadi practice as superstition and malpractice and have focused on community awareness through education, community mobilisation and mass campaigns for its elimination. However, these documents neither have the commitment of adequate financial resources nor an efficient accountability system of implementation. For instance, there are no financial resources for implementing community awareness activities as suggested by the policies formulated by provinces and local governments. Additionally, such policies lack the provision of a public reporting system, independent monitoring mechanism and remedial actions for any non-compliances.

**Lowest policy documents (e.g., directives, strategies, and action plans).** A directive was endorsed to eradicate the Chhaupadi custom in 2008 [47]. This directive had immediate and long-term interventions for preventing Chhaupadi practice. The immediate interventions included public awareness and health education, whereas the long-term interventions consisted of women empowerment through women's participation and legal reforms that ensured women's rights. Although this directive urges the district and local level agencies to implement listed interventions, however, is silent about the financial resources required for implementing those activities [47].

## Principles of equity and rights

We investigated the extent to which principles of human rights, social equity, and justice are articulated in the prevailing policy documents. Although the constitution has guaranteed the right to live with dignity [20], there were no highest-level policy documents specific to eliminate Chhaupadi practice. Only Criminal (Code) Act (2017) is the national law that has special provisions for Chhaupadi practice; however, this law aims to criminalise those who force to stay in Chhaupadi huts rather than implementing interventions for mitigation [22]. Furthermore, no other national laws ensure access to menstrual hygiene as reproductive health rights and women's human rights.

## Discussion

This study revealed the situation of Chhaupadi practice and the policy initiatives for its eradication in Nepal. Most adolescent girls (84% of N = 221) practised Chhaupadi in the Mangalsen Municipality of Achham district. The practice was higher among the girls aged 15–17 years, whose mothers were illiterate, and those who lived in a nuclear family. These adolescent girls were facing many social problems, restricted in terms of certain foods and other daily activities. Furthermore, girls had poor hygiene practice, especially among Chhaupadi practising girls. The review of policies highlighted a lack of policy documents with the highest level of authority (e.g., Acts, Regulations), which could potentially eliminate Chhaupadi practice and protect women's reproductive health rights. The Criminal Code Act (2017) is the existing legal document with the highest level of authority, aims to criminalise the one who forces the women to stay in menstrual huts. However, this Act has no provisions to ensure the right of women to access menstrual hygiene. The LGOA (2017) urges the local governments for the implementation of Chhaupadi eradication interventions. But the institutional implementation mechanism is insufficient, including financial, monitoring, and accountable systems.

### The magnitude of Chhaupadi practice

Our study reported almost all (98.2%) adolescent girls knew about Chhaupadi, while 84% practised it. These findings were higher than past studies conducted in Achham and adjoining districts. For instance, a survey conducted in Achham [1] reported a slightly low prevalence (72%) of Chhaupadi among Adolescent girls. Consistent with the findings of this study, another study conducted in the adjoining Dailekh district [23] reported 77% of women and girls practising Chhaupadi despite 60% of surveyed girls knowing Chhaupadi was illegal [48]. However, studies conducted in other parts of Nepal have reported lower prevalence [7, 29, 49]. The possible reasons behind the high prevalence in our study could be attributed to the religious and cultural values of Chhaupadi in western Nepal, especially in the Achham district [1, 23]. Furthermore, past media reporting of at least 13 deaths in the last 15 years from Achham district alone revealed how Chhaupadi huts are prevalent, and women and girls are being forced to stay there during their period [50]. Also, the etymology of the word Chhaupadi and

its origin [1] indicates that Chhaupadi practice is a deeply-rooted cultural belief in the Achham district. People believe that breaching such cultural practices can result in possible social ostracization or possible misfortune/ bad luck to the family [6, 8, 19, 51].

Several demographics, sociocultural, religious, and societal factors influence Chhaupadi practice, including the age of adolescent girls, maternal education, and type of family. In our study, adolescent girls from all backgrounds were found practising Chhaupadi. The present study revealed that daughters of illiterate mothers and girls belonging to the nuclear family were more likely to practice Chhaupadi than their counterparts; however, caste/ethnicity did not show any association. A plausible explanation could be that an illiterate mother might have poor awareness of the menstruation process and its biological mechanism, which might force adolescent girls to practice those harmful practices. Furthermore, such illiterate women could have great belief in social taboos and superstition, which might largely push them to follow such practices and pass them to new generations [52]. In addition, girls living in nuclear families might be closely observed by their parents during the menstrual and postnatal periods. Hence, the girls might continue to practice it due to fear and family pressure [53]. However, our finding contradicts another study [54]. Inconsistent with the current study, evidence from other parts in Nepal [7] and India [55] showed an association between caste/ethnicity and menstrual practice. In the current study, despite being adolescent girls (Dalits, Janajatis and Brahmin/Chhetri), almost all of them had followed the Hindu religion. As Chhaupadi has some religio-cultural misinformation of the Hindu religion, especially in western Nepal where girls are considered as impure and sins during routine period [56], therefore, are forbidden in many social, cultural, and daily activities [12, 13].

Adolescent girls faced restrictions on performing household chores, including cooking meals, touching male members, and attending cultural and religious ceremonies. Similar prohibitions on food, milk and dairy products were reported despite the need for nutritious diets during the menstruation period [57]. The reasons for restrictions could be the fear that cattle would not produce milk if menstruating girls consume dairy products [1, 58]. Adolescent girls may have some fear of being unwell if they touch anyone during the menstruation periods. Previous evidence revealed women could cause harm if they touch any objects or attend any cultural ceremonies during the menstrual period [5,12]. Previous studies from other settings of Nepal [7, 17, 32, 46, 55, 56] and India [59, 60] reported similar prohibitions during mensuration.

Our study identified hygiene practice and sanitary pads use to be low among adolescent girls. The main reasons for low utilisation could be poor access to and the high cost of sanitary pads. These findings are consistent with another study conducted in the Achham district [1]. In the current study, girls were prohibited from entering the house during their menstrual period. Likewise, in a previous study, girls were forced to do more laborious chores like carrying heavy loads, digging, fetching firewood, and collecting grass [12]. Such physical activities without providing a nutritious diet indicate an extreme form of violence against women, violation of their self-dignity and their reproductive health rights [5].

## Policies on eradication of Chhaupadi practice in Nepal

There were some policies and strategies for eliminating the Chhaupadi practice in Nepal. However, several factors were contributing to policy comprehensiveness, effective means of implementation that could cover human rights and the issues of equity.

**Comprehensiveness.** Discussing the comprehensiveness of policy documents, we did not find any Chhaupadi-specific policy formulated in a holistic approach. Furthermore, we did not find any gold standard literature that explains a comprehensive strategy to eradicate the

social custom-like Chhaupadi. However, evidence shows that eradicating other harmful practices like child marriage requires interventions that integrate legal efforts and other supportive interventions. Such comprehensive interventions include empowering girls, educating and mobilising parents and community members, supporting girls for enrollment and continuation in schools and offering economic supports and incentives [61, 62].

In Nepal, Chhaupadi eradication policies are implemented on an ad hoc basis. Most of them are coercive and aim to demolish the menstrual huts or criminalise the accused person without adequate supportive enforcement strategies [6]. For instance, more than 9,210 menstrual huts in 19 districts of Karnali and Sudurpaschim provinces were demolished through a campaign in 2020. More than 6,146 menstrual huts were demolished in the Achham district alone [63]. The Chhaupadi huts were demolished without providing alternative solutions resulting in women being forced to stay in the cowshed during the menstrual period [63]. Although new menstrual huts have been stopped, women are forced to remain in more vulnerable places like cowsheds after demolition.

Non-acceptance by parents-in-law and the psychological fear of getting ill are the reasons behind the prevailing Chhaupadi practice [63]. The existing eradicating approaches, like the demolition of menstrual huts, are not effective. The evidence from the Achham district suggests that more than one-fifth of menstrual huts have been reconstructed again after the demolition campaign [63]. Thus, a multilevel/ multidisciplinary and comprehensive policy framework only could address individual, family, community, and societal barriers for Chhaupadi eradication. Such a comprehensive framework can be designed to adopt a socio-ecological model [62]. It must be embedded in policies of all levels to ensure the right of protection, health, and survival of every woman and girl. Policy interventions need to be targeted and focused on affected communities and individuals through formal/informal systems that may influence to eradicate this centuries-long harmful cultural practice.

**Effective means of implementation.** Our analysis showed that the Criminal Code Act (2017) [23] is only the policy with the highest level of authority to eradicate Chhaupadi practice. Yet, there are several challenges in its effective implementation. Firstly, it criminalises the one who forces the women to stay in Chhaupadi huts. This person will be no other than a close family member. Filing a criminal charge against own family member could break the social fabric and family relationship.

Furthermore, administrative measures such as arresting the accused person by police and filing a case in district court are even challenging due to over-complicated bureaucratic processes. Likewise, Chhaupadi is a complex and multifaceted problem that requires coordinated efforts from the administration, police, court, schools, health facilities, family, community, and civil society. Thus, implementing coercive measures like the demolition of Chhaupadi huts is ineffective unless obtaining commitments from relevant stakeholders. The governments have enforced several interventions to eradicate Chhaupadi at their local level; however, such efforts often lack concrete action plans, strategies, and adequate financial resources.

Although provincial and local governments have initiated to endorse the policies to eradicate Chhaupadi, there are no specific policies at the federal level to ensure menstrual hygiene or eradicate Chhaupadi practice. The Dignified Menstruation Policy (2019) is the only policy specific to Menstrual Hygiene Management (MHM); however, it has not been endorsed yet. The drafted policy has encouraged mobilising religious and social leaders and change agents (individual, group, or community) to eradicate menstruation superstition, accustom, and beliefs. The aforementioned drafted policy also encourages legal actions and reforms, multisectoral and multi-layer coordination to eradicate Chhaupadi practice [42]. Although the document emphasises strong inter-ministerial coordination, the delay in endorsement of this

policy was due to a lack of clarity among the sectoral ministries (Health, Education, and the Water Supply) in leadership and taking overall implementation responsibilities [48].

Many policy documents lack a public reporting mechanism, independent monitoring system, and provision of remedial actions for any non-compliance, resulting in weak policy implementation. For example, a multi-country study conducted in Africa (Mozambique, Senegal, and Tanzania) showed the lack of policy coherence, enforcement, accountability mechanisms, and adequate financing resulted in poor policy implementation [64]. Although integrated actions with clarity in roles and responsibilities, robust reporting mechanism, guaranteeing of confidentiality, and independent investigations mechanism have proven to be effective in policy implementation [64], adequate policy harmonisation is lacking at the implementation level. Moreover, the overlapping roles and responsibilities have created ambiguity in implementation.

Furthermore, our quantitative study suggests a high prevalence of social and cultural aspects of Chhaupadi practice. In contrast, qualitative findings suggest inadequate policy provisions to address those issues at the implementation level. In addition, these policies lack adolescent responsive approaches and hence have not identified evidence-based interventions that can address socially deep-rooted customs on Chhaupadi practice. The Sudurpaschim Province endorsed the Chhaupadi Malpractice Elimination Policy (2018) [43]; but, this policy lacks secured financial resources, a structured policy progress reporting system, an independent monitoring mechanism, and appropriate means for remedial actions.

A policy directive on eliminating the Chhaupadi practice (2008) also had ambiguity in different stakeholders' roles and responsibilities. Surprisingly, the directive was silent regarding the assurance of financial resources to implement those activities, indicating the directive was endorsed to fulfil the judicial obligation. Earlier, after hearing the case of Dil Bahadur Bishwakarma and others versus Cabinet Secretariat (Writ No. 7531 of the year 2005), the supreme court ordered the government to eliminate Chhaupadi practice with appropriate legal reforms and community mobilisation [47]. This argument can be substantiated because the national laws to eliminate Chhaupadi practice with appropriate legal reforms and community mobilisation were endorsed 12 years after the judicial order [65]. Importantly, quantitative findings revealed many social and cultural practices and restrictions at the ground level; these issues should be addressed by municipal level/context-specific plans and strategies that are yet to be formulated and implemented in coordination with local-level stakeholders.

**Principles of equity and rights.** The human rights approach in eliminating Chhaupadi practice aims to provide enabling environment to live with dignity, maintain privacy, ensure access to quality healthcare, quality education and the working environment in a gender transformative approach during menstrual periods [66, 67]. There is a paucity of the highest level of policy documents to protect women's dignity, self-respect, and reproductive health rights associated with Chhaupadi practice.

Forcing girls to live in Chhaupadi huts during the menstrual period violates human rights and survival rights. Further, women/girls are exposed to the risk of malnourishment, infectious diseases, gynaecological complications, psychological impacts, and snakebite [68]. Policies on eradicating Chhaupadi practice are not designed from the human rights perspective, despite Nepal being a signatory of several international conventions and treaties like the Convention on the Elimination of All Forms of Discrimination against Women [69] and the Convention of Rights of the Child [70].

## Strengths and limitations

The strengths of the study included a pretested questionnaire. Furthermore, the fourth author (GK), being a law graduate, utilised his expertise to analyse the legal aspects of the content of

policy documents. Limitations of the study included: first, the quantitative component of this study was limited to selected wards of Mangalsen municipality among the adolescent girls only due to resources and time constraints. Hence it may not be appropriate to generalise the findings in larger geographic areas or among women of all reproductive ages. Second, analysis of policy documents might not have been able to capture the stories behind the girls practising Chhaupadi or the policy loopholes during implementation. The current study identified some sociocultural and religious work restrictions during the menstrual period due to Chhaupadi practice. However, existing policies are silent towards addressing the issues at the ground level rather than implementing coercive interventions (e.g., demolishing Chhau huts) and criminalizing socio-culturally rooted problems. Future qualitative studies should be conducted to unpack these gaps focusing on why women practice Chhaupadi and explore the gaps in policy and practice. Third, this study integrated qualitative and quantitative methods at the conceptual design stage to understand the detailed perspectives of the prevalence of the Chhaupadi problem and policy provisions for Chhaupadi eradication. The discussion section interprets the findings and draws policy, program, and research implications of Chhaupadi practice. Future studies can be conducted integrating methods at the data collection and data analysis level.

## Implications for policy and programs

This study has some implications for programs and policies. Firstly, it emphasised creating an enabling environment where women can access adequate and correct information about their rights, including reproductive health rights and the right to live with dignity. As Chhaupadi practices are rooted in complex sociocultural contexts, eradication strategies should involve individuals, closest family members (male members), communities, local government, and civil society. Multilevel interventions are more effective than a single component in changing harmful traditional practices [13]. Second, legal approaches against Chhaupadi will only be effective if they create an enabling environment involving stakeholders across different levels [7]. So, local governments can formulate specific policies that can engage the community and the key stakeholders to implement, monitor, and evaluate concrete action plans. Third, this study illustrated the need for multisectoral actions to raise awareness through different platforms (e.g., health facilities, schools, mass media, and social media). Finally, adverse mental conditions have been experienced by women and girls in addition to reproductive health problems and inadequate menstrual hygiene. Thus, mental health needs to be prioritized and services requires to be provided through psychosocial counsellors and trained health service providers.

## Conclusions

This study shows a high proportion of adolescent girls' practice Chhaupadi in western Nepal. The prevalence was high among the girls aged 15–17 years, daughters of illiterate mothers and belonging to nuclear families. Girls have poor hygiene practice during the menstrual period and restrictions in daily personal and family activities. Several policies and programs have been designed and implemented to address this practice; however, these policies are poorly implemented. Poor policy implementation is due to the lack of comprehensive interventions, inadequate and secured financial resources and a fragile accountability system. Thus, it requires formulating policy instruments with strong implementation authority, identifying appropriate interventions, and approaching multilevel stakeholders. For example, the possible eradication strategies could be community-based health awareness intervention using Social Behavior Change Communication tools and engaging the wider population, stakeholders such

as parents, grandparents, priests, teachers, traditional healers, and the local government authorities. Furthermore, allocating adequate resources and adopting a solid accountability system are equally important.

## Supporting information

**S1 Fig. Chronological policy trajectory of women rights and eradication of Chhaupadi in Nepal, 2020.**
(TIF)

## Acknowledgments

The authors would like to acknowledge the District Health Office, Achham, Mangalsen Municipality and all the participants who participated in this study.

**Disclaimer**: Views presented in this article are solely those of the authors, and do not represent views, interest, or funded work of the organisations where authors affiliated.

## Author Contributions

**Conceptualization:** Dipendra S. Thakuri, Roshan K. Thapa, Resham B. Khatri.

**Data curation:** Dipendra S. Thakuri, Roshan K. Thapa, Samikshya Singh, Geha N. Khanal, Resham B. Khatri.

**Formal analysis:** Dipendra S. Thakuri, Roshan K. Thapa, Geha N. Khanal, Resham B. Khatri.

**Methodology:** Dipendra S. Thakuri, Samikshya Singh, Geha N. Khanal, Resham B. Khatri.

**Supervision:** Resham B. Khatri.

**Writing – original draft:** Dipendra S. Thakuri, Resham B. Khatri.

**Writing – review & editing:** Dipendra S. Thakuri, Roshan K. Thapa, Samikshya Singh, Geha N. Khanal, Resham B. Khatri.

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
