## [Decision Letter · Decision Letter 0]

26 Apr 2021

PONE-D-21-07352

Menstruation taboos (Chhaupadi) practices in Western Nepal: Magnitude and existing laws and policies for eradication

PLOS ONE

Dear Dr. Thakuri,

Thank you for submitting your manuscript to PLOS ONE. After careful consideration, we feel that it has merit but does not fully meet PLOS ONE’s publication criteria as it currently stands. Therefore, we invite you to submit a revised version of the manuscript that addresses the points raised during the review process.

**Both Reviewers found the manuscript appreciable, although they noted several limitations that should be resolved before publication. Therefore, I invite the authors to address all of the Reviewers' comments.**

**In addition, I invite the Authors to use a professional English language proofreading service if one of the Authors does not declare to be a native English speaker and to have carefully revised the text.**

We look forward to receiving your revised manuscript.

Kind regards,

Stefano Federici, Ph.D.

Academic Editor

PLOS ONE

Additional Editor Comments:

Both Reviewers found the manuscript appreciable, although they noted several limitations that should be resolved before publication. Therefore, I invite the authors to address all of the Reviewers' comments.

In addition, I invite the Authors to use a professional English language proofreading service if one of the Authors does not declare to be a native English speaker and to have carefully revised the text.

Journal Requirements:

2. You indicated that you had ethical approval for your study. In your Methods section, please ensure you have also stated whether you obtained consent from parents or guardians of the minors (<18) included in the study or  whether the research ethics committee or IRB specifically waived the need for their consent.

3. Please include additional information regarding the survey or questionnaire used in the study and ensure that you have provided sufficient details that others could replicate the analyses. For instance, if you developed a questionnaire as part of this study and it is not under a copyright more restrictive than CC-BY, please include a copy, in both the original language and English, as Supporting Information.  If the original language is written in non-Latin characters, for example Amharic, Chinese, or Korean, please use a file format that ensures these characters are visible.

Reviewers' comments:

Reviewer's Responses to Questions

**Comments to the Author**

1. Is the manuscript technically sound, and do the data support the conclusions?

Reviewer #1: Yes

Reviewer #2: Yes

2. Has the statistical analysis been performed appropriately and rigorously? 

Reviewer #1: Yes

Reviewer #2: Yes

3. Have the authors made all data underlying the findings in their manuscript fully available?

Reviewer #1: Yes

Reviewer #2: Yes

4. Is the manuscript presented in an intelligible fashion and written in standard English?

Reviewer #1: Yes

Reviewer #2: No

5. Review Comments to the Author

Reviewer #1: The manuscript is well-written and informative. It addresses research questions, provides adequate data set and analysis, and I have no problems with any of the conclusions the authors have come regarding the study. It is commendable that authors have shown their hard work through the research in very rural areas of Nepal where the serious problem of the chosen topic exists. It will lend a hand for generalization in rural setting and will be helpful literature for other researchers in similar topic.

However, I would like to suggest some changes the authors could make.

1. In line 33, I would suggest you to correct the line “a century-old harmful cultural practice” as chhaupadi has been rooted for centuries in our country and it is not consistent with your other descriptions about chhaupadi in “Introduction section”.

2. Some grammar and sentence structure throughout the article is a bit off, so I would like to suggest you to use grammar corrector for efficient reading or you can go through the whole article and correct it with some effort.

3. In your Methods section, please provide additional information regarding the ethical clearance you obtained for the work from Chitwan Medical College. Furthermore, in my understanding it’s mandatory to get ethical clearance from Nepal Health Research Council (NHRC), so I wanted to know why there is no information about it. (If the ethical clearance from NHRC is not necessary then please let it be as it is)

4. In table 2 under the menstrual hygiene variable it seems categories are repeated (“new cloth” and “New cloth and old wash cloth”). The answer from the respondent might have been repeated here in data analysis. Light on those categories are necessary.

5. In table 3, please check description of highest level of authority as it is not clear (Last line of the description is not clear). The sentence structure and grammar does not match in the last line. Please review it and be sure about your understanding and what you want us to comprehend from the description.

6. As this study looks into the existing laws and policies for eradication of Cahhupadi, addition of data and information regarding middle level and low level authority might help to clear out one of the objective of this study. It seems some lacking in information.

Reviewer #2: Thank you for the opportunity to review this timely and important piece of research. The study used a mixed methods approach to determine the magnitude of chhaupadi in a particular geographical area of Nepal and studied the policy context of the practice in the country. While there are many strengths of the study, there are several questions remaining and areas in which the manuscript could be improved for clarity. Below I outline some ideas for improving the manuscript for maximum impact. Overall, the manuscript requires a major revision.

Please see attachment for detailed comments.

6. PLOS authors have the option to publish the peer review history of their article (what does this mean?). If published, this will include your full peer review and any attached files.

Reviewer #1: **Yes: **Deepmala Rana Bhat

Reviewer #2: No

---

## [Author Response · Author response to Decision Letter 0]

23 Jun 2021

Reviewer #1

1. In line 33, I would suggest you to correct the line “a century-old harmful cultural practice” as Chhaupadi has been rooted for centuries in our country and it is not consistent with your other descriptions about Chhaupadi in “Introduction section”.

Author’s response: Thank you so much for your suggestion. We have corrected it as suggested in the revised manuscript. 

2. Some grammar and sentence structure throughout the article is a bit off, so I would like to suggest you to use grammar corrector for efficient reading or you can go through the whole article and correct it with some effort.

Author’s response: Thank you so much for your suggestions. We corrected flow of the language and grammatical errors throughout the manuscript. 

3. In your Methods section, please provide additional information regarding the ethical clearance you obtained for the work from Chitwan Medical College. Furthermore, in my understanding it’s mandatory to get ethical clearance from Nepal Health Research Council (NHRC), so I wanted to know why there is no information about it. (If the ethical clearance from NHRC is not necessary then please let it be as it is).

Author’s response: Thank you so much for your important comment. The Nepal Health Research Council (NHRC) usually delegates an ethical approval authority to list of institutions to provide ethical approval at the institutions level [1], IRB of Chitwan Medical College is one of them. So, we obtained approval from CMC for this study. We added following text in the revised manuscript as:

Ethical approval was obtained from the Institutional Review Board (IRB) of Chitwan Medical College (CMC), Tribhuvan University. The Nepal Health Research Council (NHRC) generally delegates ethical approval authority to academic institutions to provide ethical approval for the institutional level researches [1]. Furthermore, permission was taken from Mangalsen municipality and Achham District Health Office. We obtained written informed consent from the respondents prior to the interview.

. 

4. In table 2 under the menstrual hygiene variable it seems categories are repeated (“new cloth” and “New cloth and old wash cloth”). The answer from the respondent might have been repeated here in data analysis. Light on those categories are necessary.

Author’s response: Thank you so much for pointing out this. It was overlooked. We again revisited the interview questionnaire and found that categories for this variable in the revised manuscript were: sanitary pad, new cloths, homemade sanitary pad, and old washed cloths. 

5. In table 3, please check description of highest level of authority as it is not clear (Last line of the description is not clear). The sentence structure and grammar does not match in the last line. Please review it and be sure about your understanding and what you want us to comprehend from the description.

Author’s response: Thank you so much for your comment, we have reviewed the description in the table and rephrased it. 

6. As this study looks into the existing laws and policies for eradication of Chhaupadi, addition of data and information regarding middle level and low-level authority might help to clear out one of the objectives of this study. It seems some lacking in information.

Author’s response: Thank you for your important suggestions. We have reviewed and added more information regarding middle and low-level authority in the revised manuscript. We have also included one separate column in the table 3 and added specific description about provision of Chhaupadi for each level of policy documents. Please refer table 3 of the revised manuscript.

Reviewer #2

1.The two key study components, 1) the quantitative survey and 2) the policy review, feel like two separate studies. If being presented in the same manuscript, the authors should consider building a stronger justification for why these two areas in particular were studied, how the results of the two components build on/support each other (mixed methods integration) and discuss linkages between the results from both components. As it reads now the two pieces feel separate. Figure 1 – Integration of quant and qual data- more detail required regarding how the data was integrated (see comment above about mixed methods integration and justifying this choice). A justification statement about the need for and use of mixed methods would add strength to the methods section. Furthermore, a discussion on methods integration would add merit, as it is mentioned in the figure but not discussed in the text. See Fetters and Creswell’s work on mixed methods integration at various states in the research design (https://www.ncbi.nlm.nih.gov/pmc/articles/PMC4097839/) 

Author’s response: Thank you so much for your insightful comment. We used both (quantitative and qualitative ) methods where the purpose of quantitative survey was to identify the prevalence of Chhaupadi while quantitative method was used to complement and expand the scope of the research gaps [2]. Addressing Chhaupadi requires an in-depth understanding of the problem and enabling policy environments and its effective implementation [3]. Thus, we used multiple methods to identify the recent prevalence and eradication policy initiatives. We have also taken reference of similar past study for design of our study [4] We integrated the quantitative and qualitative study at the conceptual level, firstly, we investigated the magnitude of the problem using quantitative survey, while qualitative study provided the macro-perspective on the policy environment to tackle this harmful religio-cultural practice of normal physiological/biological process of women/girls in our study. In addition, we interpretated quantitative and qualitative findings at the discussion stage linking findings of both methods. Therefore, this study has integrated methods at the conceptual level and findings at the discussion/interpretation stage using multi method findings.

We added a paragraph on the justification for the need of quantitative and qualitative methods for this research. Added text on the methodological justification is as:

We used quantitative and qualitative methods to identify the magnitude of Chhaupadi and review the contents of policies on Chhaupadi eradication respectively [Figure 1]. The purpose of the quantitative survey was to identify the recent magnitude of the problem, while the quantitative method was used to complement and expand the scope of the research gaps [2]. Thus, by taking reference of similar past study [4], we used multiple methods to identify the recent prevalence and eradication policy initiatives. A cross-sectional survey was conducted using face-to-face interview among adolescent girls and content of existing policies were reviewed with the focus on Chhaupadi eradication. Addressing Chhaupadi requires an in-depth understanding of the problem and enabling policy environments and their effective implementation. We integrated the quantitative and qualitative methods at the conceptual level. Firstly, we investigated the magnitude of the problem using a quantitative survey, while qualitative study provided the bigger perspective of the study. In addition, we discussed the quantitative and qualitative findings to draw the policy/program and research implications.

We totally agree with the reviewer, and literature on the mixed methods design, mixed-methods integration should be conducted at the methods, and analysis level (result) [5–7]. As this study used data of two different sources (primary- adolescent girls, and secondary- document review), methodological integration at the methods and analysis stages was not practically possible. Researchers could conduct mixed-methods research design and integrate methods at the data source and analysis level, these limitations could be the topic for further research implication. Revised text in the limitation subsection is as:

………………………Third, this study integrated qualitative and quantitative methods at the conceptual design stage to understand the detailed perspectives of the prevalence of Chhaupadi problem and policy provisions for Chhaupadi eradication and the discussion section to interpret the findings and draw policy, program, and research implications of Chhaupadi practice. Future studies can be conducted integrating methods at the data collection and data analysis level.

2.P4 Line 74/75 is not clear (grammar/language issues). 

Author’s response: Thank you for your suggestion. We have reviewed this section and corrected grammar and rephrased this section in the revised manuscript. 

3.There are differing discussions in the literature regarding the translation/definition of the root words of “chhaupadi.” See NFCC, Bist, Baumann regarding definitions of chhaupadi. Chhau refers to a “woman’s condition as untouchable” and padi refers to a state of “becoming or being.” 

o Bist BS. The effect of religious hazards in health among menstrual women : A case of far-west Nepal. Korean J Public Heal. 2014;9.

o Nepal Fertility Care Center. Assessment Study on Chhaupadi in Nepal: Towards a Harm Reduction Strategy. Kathmandu, Nepal; 2015.

o Baumann, S., Lhaki, P., Terry, M., Sommer, M., Cutlip, T., Merante, M., & Burke, J. (2021) Beyond the Menstrual Shed: Exploring Caste/Ethnic and Religious Complexities of Menstrual Practices in Far-West Nepal. Women’s Reproductive Health.

Author’s response: Thank you so much for suggestion and sharing relevent papers. We have reviewed and cited these papers. Definition of the Chhaupadi has been revised by taking reference of suggested papers. The added text on the definition of Chhaupadi is as: 

The word Chhaupadi is derived from a local Raute dialect in the far-west where "Chhau" means untouchable or unclean, and "Padi" means being or becoming [3,8,9]. Thus, Chhaupadi refers to a state of being untouchable/unclean. This harmful practice is deeply-rooted and has been practiced for centuries [10].

4. Updated literature search regarding menstrual restrictions would be beneficial for the reader (i.e., P4 lines 81-81, Amatya (in reference list but not cited here), Baumann 2021 (not in reference list, see above))

Author’s response: Thank you so much for your suggestions. We have cited recent literature in the revised manuscript. We have cited both the suggested papers. 

5. For prevalence data, it would be more appropriate to cite the direct source of a research study, rather than the UN field bulletin. For example, P4 line 110 about Chhaupadi prevalence in Accham, the UNICEF MICS would be a more appropriate and rigorous source. 

Author’s response: Thank you so much for your valuable suggestion. We have cited MICS (2019) finding; however, it only provides province wise prevalence of Chhaupadi. So, we have also cited some other district specific small size study [11] for Chhaupadi prevalence in Achham. 

6. Considering the focus of this research on a policy review, a more extensive discussion of the chhaupadi policy history would be beneficial in the introduction. For example, at the end of the introduction, the authors mention policy loopholes, but a stronger discussion and more detailed information about the policies would help the reader understand the limitations in the current policy frameworks. Contextual discussion around the current Criminal Code 2074 is needed.

Author’s response: Thank you so much for your insightful suggestion. We have reviewed and added more discussion on Chhaupadi policy history. The added text on policy history is as: 

There are many policy documents that are associated with eradication of Chhaupadi. The Constitution of Nepal (2015) ensures the right to equality (Article 18) and right to reproductive health (Article 38). Likewise, Article 24 (1) and Article 29 (2) affirm that "no one shall be treated with any kinds of untouchability or discrimination and no one shall be exploited on the basis of any custom, tradition, culture and practices or any other bases" [12]. Earlier, the Supreme Court outlawed Chhaupadi as malpractice in May 2005. The court further directed the government to take necessary legal arrangements to eliminate Chhaupadi. Later, the Government of Nepal formulated the directives to eliminate Chhaupadi practice in 2008 [13]. Thereafter, many programs such as awareness program with community stakeholders and Chhaupadi sheds demolition campaign have been implemented [9], aiming to eradicate Chhaupadi [14]. Recently, the Criminal Code (2017) criminalizes Chhaupadi and has included the provision of a three-month jail sentence and/or NPR 3,000 (~USD 26) fine for anyone forcing a woman to follow the custom [15].

7. Definition of the Criminal Code is incomplete. Fines are also a component. Ensure that the code is defined accurately and in full in the introduction. 

Author’s response: Thank you so much for your important suggestion. We have added the accurate definition of the Criminal Code Act (2017) and added the information about punishments provision. The added text on the definition of Criminal Code Act (2017) is as: 

The Criminal Code (2017) criminalizes Chhaupadi and has included the provision of a three-month jail sentence and/or NPR 3,000 (~USD 26) fine for anyone forcing a woman to follow the custom [15].

Methods 

8. Justify focus on adolescent girls specifically (and why not women as well?). Background on adolescent health should also be added to the introduction to prime the reader on this decision.

Author’s response: Thank you so much for your suggestions. We have added the text on the justification for choosing adolescent as: 

Adolescence period is the critical stage of habit formation, and most adolescents' girls experience puberty and the first mensuration in their early adolescence. Adolescents usually follow what guardians and society instruct them to do on their health and hygiene and construct their behaviour. So, we decided to recruit adolescent girls for this study.

9. Justify the focus on Achham district, and justify the purposive selection of Mangalsen? Why not select randomly, as was done for the wards? How was the list of all households with adolescent girls compiled/accessed? 

Author’s response: Thank you so much for your valuable comments. We added the text in the revised manuscript as:

As available literature suggests, out of seven provinces in Nepal, Chhaupadi is mostly prevalent in Karnali and Sudurpaschim provinces [13,16,17]. Recently many incidents, including deaths in Chhau huts were reported in media over the last decade, especially from Achham district [17–19]. Such incidents indicate that Chhaupadi is still prevalent in Achham district; therefore, we selected the Achham as a study district. Within the district, Mangalsen municipality was selected purposely out of ten local municipalities considering feasibility, time factor, accommodation, and available resources for data collection.

10. For the question, “person forcing to stay in chhaupadi huts?” was “self” an option? If not, why not? Or “other”? The question does not appear to have an exhaustive list of response options (if this cannot be addressed, it should be noted as a limitation). The same comment goes for “factors for chhaupadi practice.”

Author’s response: Thank you so much for your valuable suggestion. We have again revisited the interview questionnaire prepared in Nepali and found that the current translation in English is person suggesting to stay in Chhaupadi huts. In Nepal, Girls usually follow societal norms and values at the family level. They often follow what their parents and elder family members suggest. It is stronger in the case of religio and cultural rituals and practices. Also, Nepalese girls are not so empowered like western girls in terms of taking self-decisions and the situation is even worse in the rural hilly areas of the country. So that we have not included self-option under this variable. 

11. Variable, “types of work restriction” appears to include items that are not work related, like wearing new clothes and taking medicine. Please rectify. Variable, absorbents, new clothes is mentioned twice. Work related restriction – response options do not all appear to be about “work.”

Author’s response: Thank you so much for your important suggestions. During Chhaupadi, women and girls are often restricted to perform several daily activities such as cooking, touching others, wearing new clothes, and buying and touching medicines etc. So, here we wanted to capture the activities related restrictions during Chhaupadi practice. We have revisited questionnaire in Nepali and corrected the English translation as: “types of activity restrictions” in the revised manuscript. Regarding repetition of new clothes under types of absorbents variable, It was overlooked from us. We again revisited the interview questionnaire and found that categories for this variable in the revised manuscript were: sanitary pad, new cloths, homemade sanitary pad, and old washed cloths. 

12. Were participants compensated in any way for their participation?

Author’s response: Thank you for your query. Participation of the study respondents was voluntary. The respondents could refuse the interview process at any time. There was no provision of any in-kind or monetary support for the respondents. We appreciated all the respondents at the end of interview for their participation. 

13. Modifications regarding local dialect are mentioned, but then authors state the interviews were conducted in Nepali. Please clarify. Was the survey conducted on a tablet or pen and paper?

Author’s response: Thank you so much for your important comments. Yes, we have conducted interview in Nepali language. After pretest, we have made necessary modification especially in the flow of questions patterns and Nepali language style. So, there was no further modification of tools in local language. The survey was conducted using pen and paper. We have added text in the method section as: 

For quality assurance, survey tool was pretested among 20 adolescent girls of the adjoining ward (ward no 8) within Mangalsen municipality. Necessary modifications were made especially in the flow of the pattern of questions and language style. The second author (RKT) conducted an interview using the revised questionnaire. Interview was conducted in Nepali language using pen and paper. 

14. Policy-cube framework requires a citation. Qualitative study documents should be listed. (P8 line 210). 

Author’s response: Thank you so much for your suggestions. We have cited paper for policy cube framework. Furthermore, Qualitative study documents (e.g., constitution, laws, regulations, policies, plans, strategies, directives, judicial orders) have been listed in the revised manuscript. 

15. Figure 2 “Web searching” language is too vague to be helpful. It would be useful to know the search strategy, such as databases searched and keywords used, time the search was conducted. 

Author’s response: Thank you so much for your important suggestions. We added following text in the revised manuscript as:

We identified relevant policies through web searching and consulting with some known person in the Chhaupadi eradication movement in Nepal. We performed documents search using keywords (e.g., Karnali and Sudurpaschim province, harmful Chhaupadi practice, religio-cultural practice, practice, Chhau huts, Chhaupadi eradication, Chhaupadi practice, menstrual hygiene policy and Nepal). The documents search was conducted in June 2020. 

16. Authors state that the policy cube framework was modified for the study. How so? Author’s state they “coordinated with relevant stakeholders.” Do they mean “consulted”? Further, it would be helpful to know which stakeholders, and justify the choices made. For “documents identified” add “Policy,” as this document review was limited to policy documents, correct?In the final selection box, it would be helpful to state the number that were included in the final review. 

Author’s response: Thank you so much for your insightful suggestions. We have not modified the policy cube framework so; this might be typo error. We have corrected it. For the policy review, yes, author consulted with the relevant stakeholders. Authors identified some stakeholders considering their relevant experience in Chhaupadi eradication movement and shared our intention to review policies and request them to name the relevant policies on Chhaupadi eradication. Yes, this document review was limited to policy documents. We have added total number of policy documents included for the final review in the final selection box.

17. The authors state 84% practiced chhaupadi on their last menstruation, then state 94% practiced chhaupadi, which is confusing. Please clarify how the second question about practicing chhaupadi was asked. 

Author’s response: Thank you so much for your suggestion. The overall prevalence of the Chhaupadi practice is 84%. However, there is higher prevalence of Chhaupadi among specific age and ethnic groups compared to their counterparts. We have added revised text in the result section as: 

Out of total adolescent girls, most (84%) practised Chhaupadi during their last menstruation. Over half (56.1%) of girls were between 15-17 years of age. Three-fourth were from advantaged ethnic groups. More than half (58.8%) had completed secondary education. Four in five (81%) respondents' mothers were unpaid workers. A substantial proportion (93.5%, n=124) of Chhaupadi practice was among 15-17 years of age compared to other age groups. The daughters of illiterate mothers practised Chhaupadi more (90% of n=126) compared to mothers with a secondary level of education (75%, n=24). The practice was higher among the girls living in a nuclear family (89%, n=133) compared to joint family (77%, n=88) (Table 1).

18. Dalit and Janajati are distinct ethnic groups with quite different menstrual practices in much of Nepal, based on what is known in the literature. Thus, it was surprising to see them combined for the quantitative analysis. Please justify this choice and discuss limitations of doing so. 

Author’s response: Thank you for your suggestion.

We grouped ethnicity based on the Government of Nepal’s caste/ethnicity categorisation. For the reporting in the routine Health Management Information System [20], the Government of Nepal has categorised 123 ethnicities into six broader categories [21]: i) Dalits (Hill and Terai), ii) Janajati (disadvantaged indigenous Hill and Terai caste group), iii) Madhesi ( non-Dalit Terai caste groups), iv) religious minorities (Muslims), v) relatively advantaged indigenous groups (Brahman/Chhetri), and vi) other upper caste groups. Generally, first four groups (i, ii, iii, and iv) are considered as disadvantaged ethnicities while later two groups (v and vi) are advantaged ethnicities.

Based on comparative privilege and taking reference from previous studies [22,23], we have divided ethnicity into two groups merging Janajati with Dalit in one group and Brahman/Chhetri cast groups in another one for our study. 

Similarly, another reason is that there were Brahman/Chhetri, Dalits and Janajaties in our study and almost all of them followed Hindi religion. So Chhaupadi as religio-cultural practice based on the Hindu religious root, we merged janajati with Dalit ethnic group.

19. The way the results are discussed is confusing and contradictory at times, the results section should be revised for clarity in presentation of findings. For example, P11 line 275 states 84.8% practice chhaupadi, but at the end of the paragraph on line 281 the authors state 88.8% practice chhaupadi. This is either an error or the variables are not defined clearly in the text. Please rectify.

Author’s response: Thank you so much for your suggestions. We reviewed and revised it. 

20. Table 2 “Factors associated with chhaupadi” – Can the authors be more specific on what this is measuring? Are these reasons for why it is practiced? Clarity is necessary. If this is asking the question, “Why do you practice chhaupadi?” 4 of the response options seem plausible but the response “lack of education” seems more like a demographic variable that could be compared with chhaupadi practice, rather than a response to this question in the survey. Did respondents say they practice because they are uneducated? 

Author’s response: Thank you for bringing this into our attention; as well as allowing us to clarify this in more details. Though there is variable as lack of education, we wanted to capture knowledge as a factor of Chhaupadi practice. Respondent said they practice Chhaupadi because they were unaware and have no knowledge. We have again revisited interview questionaries in Nepali and corrected English translation of this variable as “inadequate knowledge” in the revised manuscript. 

21. Some of the variables are non-exhaustive. 

o “Bathing in public sources of” appears to be incomplete.

Author’s response: Thank you so much for your suggestions. We have reviewed this. The complete name of variable was “Bathing in public source of water” but some part was missed out due to formatting issue of the table. We have addressed this.

22. Figure 3 Readability/formatting can be improved. Full information about each policy/act is required. For example, 1996 simply states “Mental Health Policy,” which is not enough detail for the reader to find and review the document. Similarly, criminal code for 2017, the full details should be provided so reader is able to find more information as needed. 

Author’s response: Thank you for your suggestions. We have reviewed and addressed this in the revised manuscript. 

23. General comment is that there are many tables and figures, some with overlapping information. Consider reducing/combining where possible. For example, Table 3 and figure 3 could likely be combined, as there is overlapping information. Formatting of table 3 could be improved (capitalization, adjusting width of columns, etc. to improve readability)

Author’s response: Thank you for your suggestions. We have reviewed formatting of the table 3. Also, as per your suggestion, we have kept original figure 3 as supplementary figure.1 to avoid overlapping information. 

24. Consistently provide both Nepali year and Gregorian year throughout manuscript. P14 lines 317-319 seems better suited for the introduction section, not the results. Criminal code details P15 lines 355-358- should specify 3 months imprisonment and/or a financial penalty.

Author’s response: Thank you for your suggestions. We have reviewed and addressed this in the revised manuscript. We have added text as: 

The national law that had the provision of Chhaupadi practice is the Criminal (Code) Act (2017). This Act has the provision of three months' imprisonment and/ or a financial penalty of NPR 3,000 (~USD 26) for those forcing a recently delivered or menstruating woman to stay in Chhaupadi huts [15].

26. Discussion: Prevalence – Authors state that other studies found lower prevalence rates. It would be helpful if the authors provided some discussion on why it may be the case that their study found such a high prevalence of the practice compared to other studies. Is there something about this particular area or period in which the study was conducted that may be unique? 

Author’s response: Thank you so much for your important comments. We added following text in the revised manuscript as:

The possible reasons behind high prevalence in our findings could the religious and cultural values of Chhaupadi are deeply-rooted in that society[17,24]. For example, at least 13 deaths in the last 15 years were reported from Achham district alone due to forceful stay in Chhaupadi huts [25]. Also, there are beliefs that the tradition of Chhaupadi may have originated in Achham since the word Chhaupadi is derived from the local Raute language [17]. 

27. The results indicate a higher proportion of Dalit and Janajati practicing chhaupadi- how does this relates to what is known in the literature, and why this may this be the case in Achham context? It is interesting that caste/ethnicity did not play a role in chhaupadi practices, as other studies have found caste to be a significant predictor of menstrual practices in Nepal (Baumann et al. 2019) and India (Khanna et al. 2005). It will be important for the authors to compare their findings against other studies in health and menstruation that found caste/ethnicity to be a key variable and discuss why their findings may differ. 

Author’s response: Thank you for your important comments. We added following text in the revised manuscript as:

In our study, almost all adolescent girls were from Khas/Arya and Adivasi ethnic group (Dalits, Janajaties and Brahmin/Chhetri), and all of them had followed the Hindu religion. Chhaupadi has been in practice based on the Hindu religio-cultural practices in western Nepal where some women and girls are considered as impure and sins [26] during their mensuration, therefore are forbidden in many social, cultural, and daily activities [27,28]. 

27. Nuclear vs joint family This is an interesting finding and glad to see that the authors contrasted these findings against another study that found the opposite. Comprehensiveness – p20 – While the authors are correct in stating that there is no gold standard in the literature for addressing chhaupadi specifically, authors could consider reviewing/comparting policies to actions that have been taken to address other harmful practices in Nepal and globally. For example, see dowry, child marriage etc. in UNFPA study: https://nepal.unfpa.org/en/publications/literature-review-harmful-practices-nepal

Author’s response: Thank you so much for your valuable comments. We have reviewed literature and added following text in revised manuscript as:

Evidence shows that eradication of other harmful practice like child marriage requires interventions that integrate legal efforts along with other supportive interventions. Such comprehensive interventions include empowerment of girls, educating and mobilising parents and community members, supporting girls for enrollment and continuation in schools and offering economic supports and incentives [29,30]. 

28. It is suggested that the authors refrain from value-judgment language, and rather frame the issue in terms of health and rights violations. (e.g., P21 line 515 “bad cultural practice” could be replaced with “harmful cultural practice”)

Author’s response: Thank you so much for your suggestions. We have revised as per the suggestion. 

29. The finding regarding policy documents lacking a public reporting mechanism and independent monitoring system is a critical one. It would be helpful if the authors could expand upon this in the discussion (P22 Line 547) to discuss other policies/approaches that have worked effectively (even in context outside of Nepal if necessary) in order to frame the issue with a recommended step moving forward to creating adequate reporting mechanisms. What would/could a strong reporting mechanism look like for Chhaupadi, and what can we learn from what has been done to address similar policy challenges for other issues?

Author’s response: Thank you so much for your valuable suggestions. We have reviewed literature. We added following text in the revised manuscript as:

Most of the policy documents lack public reporting mechanism, independent monitoring system and provision of remedial actions for any non-compliance, which could result in weak policy implementation. A study conducted in African countries (Mozambique, Senegal, and Tanzania) shows that lack of policy coherence, enforcement, accountability mechanisms, and adequate financing results in poor implementation of policy [31]. Although integrated actions with clarity in roles and responsibilities, strong reporting mechanism, efficient internal reporting of any deviances or misconduct, guaranteeing of confidentiality and independent investigations mechanism have proven to be effective in the implementation of policy [32], most of the policies related to elimination of Chhaupadi are implemented on an ad-hoc basis. Moreover, there are overlapping roles and responsibilities and implementation ambiguity.

30. Author’s state they employed a pretested survey, however, it would add to the strength if they could include more information in the methods section discussing how the survey questions were developed. Were they adapted from an existing pre-tested tool? Were the survey questions pulled from literature on the topic?

Author’s response: Thank you so much for your important comments. The survey tools were developed by the research team based on an extensive literature review. Yes ,we developed survey tools by taking reference of literatures on the topic [17,33–36]. 

31. “Critical analysis and synthesis of policies” is not necessarily a strength. This is expected in a content review of policy. If there were multiple independent reviewers of the policies it could be argued that would be an added strength to increase rigor and validity, but that doesn’t seem to be the case in this study. 

Author’s response: Thank you so much for your comments. We agree with you. We have revised it as per your suggestion. 

32. Mental health is discussed here, if so, it should be supported by the results of the study. 

Author’s response: Thank you so much for your suggestions. 

33. References

7 – Link doesn’t work.

8 – Reference is incomplete. 

22 – Provide a link to the code if possible (even if it is in Nepali)

44 – Name is in all caps- not consistent with other references. 

40 – Remove “forthcoming” as this is already published. 

48 – Reference is incomplete.

Author’s response: Thank you so much for your suggestions. We have checked and corrected references in the revised manuscript. 

We would like to thank both the reviewers for their insightful comments and feedback. Thank you so much for inviting us for revision of this manuscript. 

 Dipendra Singh Thakuri on behalf of all co-authors

Reference 

1. NHRC. Health Research System In Nepal. Nepal Heal Res Counc. 2006. Available: http://nhrc.gov.np/wp-content/uploads/2017/02/Health-Research-System-Nepal-2006.pdf

2. Driessnack M, Sousa VD, Mendes IAC. An overview of research designs relevant to nursing: Part 3: Mixed and multiplemethods. Rev Lat Am Enfermagem. 2007;15: 1046–1049. doi:10.1590/s0104-11692007000500025

3. Baumann SE, Lhaki P, Burke J. Assessing the Role of Caste/Ethnicity in Predicting Menstrual Knowledge, Attitudes, and Practices in Nepal. Glob Public Health. 2019;0: 1–14. doi:10.1080/17441692.2019.1583267

4. Kohrt BA. Vulnerable social groups in postconflict settings: a mixed methods policy analysis and epidemiology study of caste and psychological morbidity in Nepal. Intervention. 2009;7: 239–264. doi:10.1097/wtf.0b013e3283346426

5. Bazeley P. Integrating data analyses in mixed methods research. J Mix Methods Res. 2009;3: 203–207. doi:10.1177/1558689809334443

6. Fetters MD, Curry LA, Creswell JW. Achieving integration in mixed methods designs - Principles and practices. Health Serv Res. 2013;48: 2134–2156. doi:10.1111/1475-6773.12117

7. Moseholm E, Fetters MD. Conceptual models to guide integration during analysis in convergent mixed methods studies. Methodol Innov. 2017;10: 205979911770311. doi:10.1177/2059799117703118

8. UNFPA, UNICEF, UNRCO. Literature Review of Harmful Practices in Nepal. 2019. Available: https://nepal.unfpa.org/en/publications/literature-review-harmful-practices-nepal

9. NFCC. Assessment Study on Chhaupadi in Nepal:Towards a Harm Reduction Strategy. 2015. Available: http://www.nfcc.org.np/2019/10/21/chhaupadi-assessment/

10. Karki, K. B., Poudel, P. C., Rothchild, J., Pope, N., Bobin, N. C., Gurung, Y., Basnet, M., Poudel, M., Sherpa LY. Scoping Review and Preliminary Mapping Menstrual Health and Hygiene Management in Nepal. Population Services International Nepal. 2017. 

11. Amatya P, Ghimire S, Callahan KE, Baral BK, Poudel KC. Practice and lived experience of menstrual exiles (Chhaupadi) among adolescent girls in far-western Nepal. PLoS One. 2018;13: 1–17. doi:10.1371/journal.pone.0208260

12. Government of Nepal. Constitution of Nepal, 2015. Kathmandu, Nepal; 2015. Available: http://www.moljpa.gov.np/en/category/constitution/

13. Kadariya S, R. Aro A. Chhaupadi practice in Nepal - analysis of ethical aspects. Medicolegal Bioeth. 2015; 53. doi:10.2147/mb.s83825

14. Dahal B.P., Acharya, S., R., Sunar, T. & Parajuli B (2017). Implementing Status of National Laws, Polices and Guideline: A study on situation of implementation status of National laws, policies and Chhaupadi elimination guideline in Karnali, Nepal. Kathmandu: Action Works Nepal & BEE Group. 2017. Available: https://www.actionworksnepal.org/publications/research-report/

15. Government of Nepal. Criminal (Code) Act, 2017 (Unofficial Translation). Kathmandu; 2017. Available: http://www.moljpa.gov.np/en/category/acts/

16. United Nations Children’s Fund, National Planning Commission. Monitoring the situation of children and women Nepal: Multiple Indicator Cluster Survey Final Report. Kathmandu, Nepal; 2014. Available: https://www.unicef.org/nepal/reports/multiple-indicator-cluster-survey-final-report-2014

17. Amatya P, Ghimire S, Callahan KE, Baral BK, Poudel KC. Practice and lived experience of menstrual exiles (Chhaupadi) among adolescent girls in far-western Nepal. PLoS One. 2018;13. doi:10.1371/journal.pone.0208260

18. National Human Right Commission. National Investigation Program Report on Chhaupadi (Unofficial Translation). Kathmandu; 2018. 

19. United Nations Resident and Humanitarian Coordinator’s Office. Field Bulletin Chhaupadi in the Far-West. 2011. Available: https://www.ohchr.org/Documents/Issues/Water/ContributionsStigma/others/field_bulletin_-_issue1_april_2011_-_chaupadi_in_far-west.pdf

20. Banstola A, Banstola A. The Epidemiology of Hospitalization for Pneumonia in Children under Five in the Rural Western Region of Nepal: A Descriptive Study. PLoS One. 2013;8: 1–5. doi:10.1371/journal.pone.0071311

21. Umesh G, Jyoti M, Arun G, Sabita T, Yogendra P, Tesfayi G. Inequalities in health outcomes and access to services by caste/ethnicity, province, and wealth quintile in Nepal. DHS Furth Anal Rep. 2019. Available: https://dhsprogram.com/pubs/pdf/FA117/FA117.pdf

22. Khanal V, Adhikari M, Karkee R, Gavidia T. Factors associated with the utilisation of postnatal care services among the mothers of Nepal: Analysis of Nepal Demographic and Health Survey 2011. BMC Womens Health. 2014;14: 1–13. doi:10.1186/1472-6874-14-19

23. Adhikari TB, Rijal A, Kallestrup P, Neupane D. Alcohol consumption pattern in western Nepal: Findings from the COBIN baseline survey. BMC Psychiatry. 2019;19: 1–8. doi:10.1186/s12888-019-2264-7

24. Adhikari R. Bringing an end to deadly “menstrual huts” is proving difficult in Nepal. BMJ. 2020;368: 1–2. doi:10.1136/bmj.m536

25. Bhrikuti Rai. Women are still dying after being sent to menstruation huts, but no one is filing complaints. The Kathmandu Post. 2019. Available: https://kathmandupost.com/national/2019/03/22/women-are-still-dying-after-being-sent-to-menstruation-huts-but-no-one-is-filing-complaints

26. United Nations Nepal. Harmful practices in Nepal: Report on community perceptions. 2020. Available: https://nepal.unfpa.org/sites/default/files/pub-pdf/Harmful Practices Perception Survey.pdf

27. Thapa S, Aro AR. ‘Menstruation means impurity’: multilevel interventions are needed to break the menstrual taboo in Nepal. BMC Womens Health. 2021;21: 1–5. doi:10.1186/s12905-021-01231-6

28. Niranjan Khadka. Chhaupadi Pratha: Women’s Condition and Suffering. Molung Educ Front. 2020;10: 81–92. doi:https://doi.org/10.3126/mef.v10i1.34031

29. Malhotra A, Warner A, McGonagle A, Lee-Rife S. Solutions to End Child Marriage. Int Cent Res Women. 2011. Available: http://www.icrw.org/files/publications/Solutions-to-End-Child-Marriage.pdf

30. Lee-rife S, Malhotra A, Warner A, Glinski AM. What Works to Prevent Child Marriage : A Review of the Evidence. Stud Fam Plann. 2012;43: 287–303. doi:doi: 10.1111/j.1728-4465.2012.00327.x.

31. Mugwagwa J, Edwards D, de Haan S. Assessing the implementation and influence of policies that support research and innovation systems for health: The cases of Mozambique, Senegal, and Tanzania. Heal Res Policy Syst. 2015;13: 1–7. doi:10.1186/s12961-015-0010-2

32. Transparency International. The business case for speaking up. 2017. Available: https://images.transparencycdn.org/images/2017_BusinessCaseSpeakingUp_EN.pdf

33. Sumpter C, Torondel B. A Systematic Review of the Health and Social Effects of Menstrual Hygiene Management. PLoS One. 2013;8. doi:10.1371/journal.pone.0062004

34. Van Eijk AM, Sivakami M, Thakkar MB, Bauman A, Laserson KF, Coates S, et al. Menstrual hygiene management among adolescent girls in India: A Systematic review and meta-analysis. BMJ Open. 2016;6. doi:10.1136/bmjopen-2015-010290

35. Mukherjee A, Lama M, Khakurel U, Jha AN, Ajose F, Acharya S, et al. Perception and practices of menstruation restrictions among urban adolescent girls and women in Nepal: A cross-sectional survey. Reprod Health. 2020;17: 1–10. doi:10.1186/s12978-020-00935-6

36. Ranabhat C, Kim CB, Choi EH, Aryal A, Park MB, Doh YA. Chhaupadi Culture and Reproductive Health of Women in Nepal. Asia-Pacific J Public Heal. 2015;27: 785–795. doi:10.1177/1010539515602743

---

## [Decision Letter · Decision Letter 1]

2 Aug 2021

PONE-D-21-07352R1

A harmful religio-cultural practice (Chhaupadi) during menstruation among adolescent girls in Nepal: Prevalence and policies for eradication

PLOS ONE

Dear Dr. Thakuri,

Thank you for submitting your manuscript to PLOS ONE. After careful consideration, we feel that it has merit but does not fully meet PLOS ONE’s publication criteria as it currently stands. Therefore, we invite you to submit a revised version of the manuscript that addresses the points raised during the review process.

** I invite the Authors to use a professional English language proofreading service if one of the Authors does not declare to be a native English speaker and to have carefully revised the text.**

We look forward to receiving your revised manuscript.

Kind regards,

Stefano Federici, Ph.D.

Academic Editor

PLOS ONE

Journal Requirements:

Additional Editor Comments (if provided):

I invite the Authors to use a professional English language proofreading service if one of the Authors does not declare to be a native English speaker and to have carefully revised the text.

Reviewers' comments:

Reviewer's Responses to Questions

**Comments to the Author**

1. If the authors have adequately addressed your comments raised in a previous round of review and you feel that this manuscript is now acceptable for publication, you may indicate that here to bypass the “Comments to the Author” section, enter your conflict of interest statement in the “Confidential to Editor” section, and submit your "Accept" recommendation.

Reviewer #1: All comments have been addressed

Reviewer #2: All comments have been addressed

2. Is the manuscript technically sound, and do the data support the conclusions?

Reviewer #1: Yes

Reviewer #2: Yes

3. Has the statistical analysis been performed appropriately and rigorously? 

Reviewer #1: Yes

Reviewer #2: Yes

4. Have the authors made all data underlying the findings in their manuscript fully available?

Reviewer #1: Yes

Reviewer #2: No

5. Is the manuscript presented in an intelligible fashion and written in standard English?

Reviewer #1: Yes

Reviewer #2: Yes

6. Review Comments to the Author

Reviewer #1: I wish congratulations to authors for their hard work and completing this research article. I was happy to review the article. And in my opinion, authors have done pretty good job in addressing the reviews and corrected every issues. So, I don't have any other comment regarding the article. This article is in line with PLOS ONE's criteria and fulfilled it.

I thank the Journal for this opportunity.

Reviewer #2: Thanks to the authors for their rigorous dedication to addressing all the comments. Just a couple pending notes for the authors:

- In reading through the manuscript I still found a number of places where the grammar could be improved.

- Citation 51 (page 23) is noted in text but not reference list. The following study supports the author's findings about family influence, but is not referred to in the discussion, add it would help to build author's argument (Beyond the Menstrual Shed: Exploring Caste/Ethnic and Religious Complexities of Menstrual Practices in Far-West Nepal - Sara E. Baumann, Pema Lhaki, Martha A. Terry, Marni Sommer,

Trevor Cutlip, Monica Merante, and Jessica G. Burke)

I hope this review was helpful and I enjoyed reading the revised version of this manuscript which will significantly advance knowledge on this topic. Thank you for all your hardwork on this study!

7. PLOS authors have the option to publish the peer review history of their article (what does this mean?). If published, this will include your full peer review and any attached files.

Reviewer #1: **Yes: **Deepmala Rana Bhat

Reviewer #2: No

---

## [Author Response · Author response to Decision Letter 1]

18 Aug 2021

Reviewer #1

1. I wish congratulations to authors for their hard work and completing this research article. I was happy to review the article. And in my opinion, authors have done pretty good job in addressing the reviews and corrected every issues. So, I don't have any other comment regarding the article. This article is in line with PLOS ONE's criteria and fulfilled it. 

 Authors response: Thank you so much for this compliment. 

Reviewer #2: Thanks to the authors for their rigorous dedication to addressing all the comments. Just a couple pending notes for the authors:

1. In reading through the manuscript, I still found a number of places where the grammar could be improved.

 Author’s response: Thank you so much for your suggestions. The final revised manuscript is edited for the flow of the language and grammatical errors. We got language and copy edits support from one of the bilingual (Nepali and English) academic as well as an associate editor of BMC Public Health. 

2. Citation 51 (page 23) is noted in text but not reference list. 

 Author’s response: Thank you so much for your comment, we have reviewed it and corrected in the revised manuscript.

3. The following study supports the author's findings about family influence, but is not referred to in the discussion, add it would help to build author's argument (Beyond the Menstrual Shed: Exploring Caste/Ethnic and Religious Complexities of Menstrual Practices in Far-West Nepal - Sara E. Baumann, Pema Lhaki, Martha A. Terry, Marni Sommer,Trevor Cutlip, Monica Merante, and Jessica G. Burke)

 Author’s response: Thank you so much for your comment, we have cited the suggested paper in the revised manuscript. 

We would like to thank both the reviewers for their insightful comments and feedback. Thank you so much for inviting us for revision of this manuscript. 

Dipendra Singh Thakuri on behalf of all co-authors

---

## [Editor Report · Decision Letter 2]

20 Aug 2021

A harmful religio-cultural practice (Chhaupadi) during menstruation among adolescent girls in Nepal: Prevalence and policies for eradication

PONE-D-21-07352R2

Dear Dr. Thakuri,

We’re pleased to inform you that your manuscript has been judged scientifically suitable for publication and will be formally accepted for publication once it meets all outstanding technical requirements.

Kind regards,

Stefano Federici, Ph.D.

Academic Editor

PLOS ONE
---

## [Editor Report · Acceptance letter]

24 Aug 2021

PONE-D-21-07352R2 

A harmful religio-cultural practice (*Chhaupadi*) during menstruation among adolescent girls in Nepal: Prevalence and policies for eradication 

Dear Dr. Thakuri:

I'm pleased to inform you that your manuscript has been deemed suitable for publication in PLOS ONE. Congratulations! Your manuscript is now with our production department. 

Kind regards, 

on behalf of

Prof. Stefano Federici 

Academic Editor

PLOS ONE